# Patient preferences and cost-benefit of hypertension and hyperlipidemia collaborative management model between pharmacies and primary care in Portugal: A discrete choice experiment alongside a trial (USFarmácia®)

**Suzete Costa**[1,2,3]*, **José Guerreiro**[4], **Inês Teixeira**[4], **Dennis K. Helling**[5], **Céu Mateus**[6], **João Pereira**[1,7]

**1** NOVA National School of Public Health (ENSP), Universidade NOVA de Lisboa, Lisboa, Portugal, **2** Institute for Evidence-Based Health (ISBE), Lisboa, Portugal, **3** Faculdade de Medicina, Universidade de Lisboa, Lisboa, Portugal, **4** Centre for Health Evaluation & Research (CEFAR), Infosaúde, Associação Nacional das Farmácias, Lisboa, Portugal, **5** Skaggs School of Pharmacy and Pharmaceutical Sciences, University of Colorado, Denver, Colorado, United States of America, **6** Health Economics at Lancaster, Division of Health Research, Lancaster University, Lancaster, United Kingdom, **7** Public Health Research Centre (PHRC/CISP), Comprehensive Health Research Centre (CHRC), Lisboa, Portugal

* suzete.costa@isbe.research.ulisboa.pt

**Data Availability Statement:** The anonymized dataset "Base DCE" generated during and analyzed

## Abstract

### Background

Little is known about patient preferences and the value of pharmacy-collaborative disease management with primary care using technology-driven interprofessional communication under real-world conditions. Discrete Choice Experiments (DCEs) are useful for quantifying preferences for non-market services.

### Objectives

1) To explore variation in patient preferences and estimate willingness-to-accept annual cost to the National Health Service (NHS) for attributes of a collaborative intervention trial between pharmacies and primary care using a trial exit DCE interview; 2) to incorporate a DCE into an economic evaluation using cost-benefit analysis (CBA).

### Methods

We performed a DCE telephone interview with a sample of hypertension and hyperlipidemia trial patients 12 months after trial onset. We used five attributes (levels): waiting time to get urgent/not urgent medical appointment (7 days/45 days; 48 hrs./30 days; same day/15 days), model of pharmacy intervention (5-min. counter basic check; 15-min. office every 3 months for BP and medication review of selected medicines; 30-min. office every 6 months for comprehensive measurements and medication review of all medicines), integration with

during the current study contains potentially identifying and sensitive patient information and is deposited in the Institute for Evidence-Based Health (ISBE) data repository. This dataset may be available for unrestricted use to researchers upon request (isbe@isbe.research.ulisboa.pt).

**Funding:** The trial was promoted by the Agrupamento dos Centros de Saúde (ACeS) [NHS Group of Primary Care Units] of Baixo Mondego Region and the Associação Nacional das Farmácias (ANF) [National Association of Pharmacies] and was funded by the ANF. ANF also funded the Discrete Choice Experiment interviews. The funder provided support in the form of salaries for authors SC, JG, and IT but did not have any additional role in the study design, data collection and analysis, decision to publish, or preparation of the manuscript. The specific roles of these authors are articulated in the 'author contributions' section. SC paid for the open-access publication fee for this manuscript.

**Competing interests:** The funder provided support in the form of salaries for authors SC, JG, and IT. DKH has given talks on US innovative pharmacist-led collaborative interventions in Portugal for which travel, and accommodation costs have been reimbursed by ANF. CM and JP declare that they have no known conflicts of interest. This does not alter our adherence to PLOS ONE policies on sharing data and materials.

primary care (weak; partial; full), chance of having a stroke in 5 years (same; slightly lower; much lower), and annual cost to the NHS (0€; 30€; 51€; 76€). We used an experimental orthogonal fractional factorial design. Data were analyzed using conditional logit. We subtracted the estimated annual incremental trial costs from the mean WTA (Net Benefit) for CBA.

## Results

A total of 122 patients completed the survey. Waiting time to get medical appointment—on the same day (urgent) and within 15 days (non-urgent)—was the most important attribute, followed by 30-minute pharmacy intervention in private office every 6 months for point-of-care measurements and medication review of all medicines, and full integration with primary care. The cost attribute was not significant. Intervention patients were willing to accept the NHS annual cost of €877 for their preferred scenario. The annual net benefit per patient is €788.20 and represents the monetary value of patients' welfare surplus for this model.

## Conclusions

This study is the first conducted in Portugal alongside a pharmacy collaborative trial, incorporating DCE into CBA. The findings can be used to guide the design of pharmacy collaborative interventions with primary care with the potential for reimbursement for uncontrolled or at-risk chronic disease patients informed by patient preferences. Future DCE studies conducted in community pharmacy may provide additional contributions.

## Trial registration

Current Controlled Trials (ISRCTN): ISRCTN13410498, retrospectively registered on 12 December 2018.

## Introduction

Stated-preference methods may quantify preferences for attributes of a medical product or intervention, using, for instance, Discrete Choice Experiments (DCE) [1]. DCEs are grounded in Random Utility Theory and Lancaster's Theory [2–4] which consider that respondents make trade-offs between service attributes.

DCEs have also been used in pharmacy-based patient care services. When cost is included, we can estimate Willingness-to-Pay (WTP) or Willingness-to-Accept (WTA) [1,5]. This method can include both the value of service as a commodity and as a public good [6].

We can then further incorporate WTP/WTA into a cost-benefit analysis (CBA) [7]. CBA requires program consequences to be valued in monetary units, it is broad in scope and able to capture health and wider benefits beyond cost-effectiveness (CEA) or cost-utility analysis (CUA) [8]. This is interesting for innovative public health interventions that may encompass wider benefits beyond health and where no real market value exists, such as pharmacy-based services with evidence of effectiveness [9,10] to inform reimbursement decisions.

In 2019, a systematic review funded by the European Commission stated that DCEs could serve as the basis for a harmonized approach to assessing public policies on new health technologies in the European Union [11]. In 2022, the European Medicines Agency issued a positive opinion on the IMI-PREFER Recommendations on why, when, and how to assess and use

patient preferences in medical product decision-making [12], for the first time, in the regulatory decision.

In 2015, a systematic review identified 17 DCE studies of pharmacy services [7]. In 2018, Dawoud reported 23 DCE studies [13]. However, not all studies elicited patients' preferences, e.g., the first DCE in pharmacy in 2002 [14].

Ten studies included in the Vass et al review [7] elicited preferences from patients: 2 looked at minor ailments and emergency hormonal contraception [15,16], 5 addressed medication therapy management or disease management [17–21], 2 were on coagulation factor [22,23], and one was on prescribing by pharmacists [24].

Since then, 12 more DCE studies were published on patient preferences for a pharmacy-delivered patient care service: 2 on minor ailments [25,26], 4 on medication therapy management or disease management [27–30], 2 on innovative pharmacy services [31,32], 1 on the Hepatitis C test [33], 1 on specialty medicines [34], 1 on screening for cardiovascular disease [35], and 1 on prescribing by pharmacists [36]. In 2016, Tinelli incorporated a DCE into an economic evaluation using CBA. WTP and costs were collected alongside a collaborative trial between pharmacists and General Practitioners (GPs) [28]. In 2019, a study on the management of high blood pressure (BP) framed cost as cost to the National Health Service (NHS) looking at a reimbursed model of care [29].

These 22 DCE studies eliciting patient or public preferences in pharmacy patient care services were conducted in the UK (10), USA (4), Australia (3), Italy (2), The Netherlands (1), China (1), and New Zealand (1).

In Portugal, a landmark study in 2008 estimated the volume, cost, and economic value of pharmacist advice on point-of-care measurements, prescription medicines, and non-prescription medicines. It appears to be the first study in pharmacy services worldwide to incorporate DCE into CBA. Three attributes were defined: counseling model; waiting time in pharmacy; and cost. The study used conditional logit regression. The WTP estimate was €76.5 M in 2008. Costs were valued at €28.4 M. The net benefit was estimated at €48.1 M [37]. However, it did not assess preferences for a collaborative patient care intervention between pharmacies and primary care.

In 2013, 12.3% of all hospital admissions in Portugal were due to Ambulatory Care Sensitive Conditions (ACSCs) and 93.7% occurred following emergency room visits. The third and fourth most common ACSCs were heart failure and hypertensive heart disease, often as a consequence of poorly managed hypertension and hyperlipidemia, which creates an opportunity to improve service delivery [38].

Planning and conducting a well-designed trial for the assessment of effectiveness and other dimensions should precede the economic evaluation of pharmacy-based public health interventions [39].

Therefore, we developed a controlled trial to assess the effectiveness [40], cost-effectiveness and cost-utility [41], and cost-benefit of the first real-world collaborative intervention in hypertension and/or hyperlipidemia management using technology-driven data exchange between pharmacies and primary care in Portugal versus usual (fragmented) care to guide future experiments with the potential of reimbursement.

This paper presents the third work of this research.

The aims of this study are: 1) to explore variation in patient preferences and estimate WTA annual cost to the NHS for attributes of a collaborative care model between pharmacies and primary care in Portugal, using a DCE alongside a trial; 2) to incorporate DCE into an economic evaluation using CBA.

This study is the first conducted in Portugal alongside a pharmacy collaborative trial, incorporating DCE into CBA.

## Methods

We followed ISPOR guidelines: ISPOR Checklist for Conjoint Analysis [5], ISPOR Report for Experimental Designs for DCEs [42], and ISPOR Report of Statistical Methods for the Analysis of DCEs [43].

### Trial-based study

A DCE was performed alongside a trial (USFarmácia® Trial) as a trial exit interview. Patients were recruited in pharmacies according to inclusion criteria: adult NHS patients on medication for hypertension and/or hyperlipidemia, either new to therapy or usual medication users, preferably with baseline blood pressure and total cholesterol above reference values. Intervention patients were also mobile phone users (to receive SMS refill reminders), consenting for data exchange between pharmacists and general practitioners, and holders of a patient record. The study intervention consisted of hypertension and/or hyperlipidemia management within a collaborative framework according to consensus-based clinical decision algorithms integrated into the pharmacy dispensing software with data exchange with primary care [40].

The trial was registered with Current Controlled Trials (ISRCTN): ISRCTN13410498.

The Administração Regional de Saúde (ARS Centro) [Regional Health Administration] approved the trial and study on 09-02-2017 following the opinion of its ethics committee, Comissão de Ética para a Saúde of ARS Centro on 01-02-2017. The ethics committee Instituto de Bioética of Universidade Católica Portuguesa also approved the study (Ethical Screening Report 02/2018) on 20-03-2018. In early 2018, we revised all data protection procedures to meet the new General Data Protection Regulation (GDPR) entering into force on 25 May 2018 and we detailed these in the Privacy Impact Assessment of the trial on 28-02-2018 which was provided to both Ethics Committees. Participants provided written consent. Consent Forms included provisions for collecting economic data and consent for the DCE study was reconfirmed at the end of the trial just before the DCE survey.

### DCE attributes and levels

Attributes and levels were informed by a mixed-methods procedure, comprising: 1) the aims and features of the USFarmácia® trial intervention arm; 2) a trial exit patient focus group using thematic analysis; 3) a review of DCE studies on pharmacy-based interventions [7,15–28,31,33] and hypertension management [29]; 4) a DCE study on pharmacy interventions in Portugal [37]; 5) patient perceptions on innovative pharmacy-based interventions in Portugal [44]; 6) patient preferences on primary care delivery in Portugal [45]; 7) the Portuguese NHS Annual Report 2018 on Access to Health Care [46]; 8) and experts' opinion (**Fig 1**).

We first selected the most frequent attributes found in DCE studies of pharmacy-based interventions: model of pharmacy intervention, waiting time, chance of best treatment, and cost [7,15–28,31,33]. These were also confirmed by patient preferences [37,44] and experts' opinion. We added integration with primary care based on the aims and features of the trial intervention arm and the patient focus group.

Levels were informed by the different sources as outlined further. Level 1 was defined as usual care in all attributes (almost no collaborative or integrated care). The remaining levels sought to illustrate possible responses within a collaborative care model. Level 2 represents improved care (vs. usual care) and level 3 represents the care level provided in the intervention arm of the collaborative care trial (the highest form of collaborative care) for all attributes except cost. Levels of the cost attribute are explained further.

The final set of attributes and levels after the pre-test is shown in **Fig 2**.

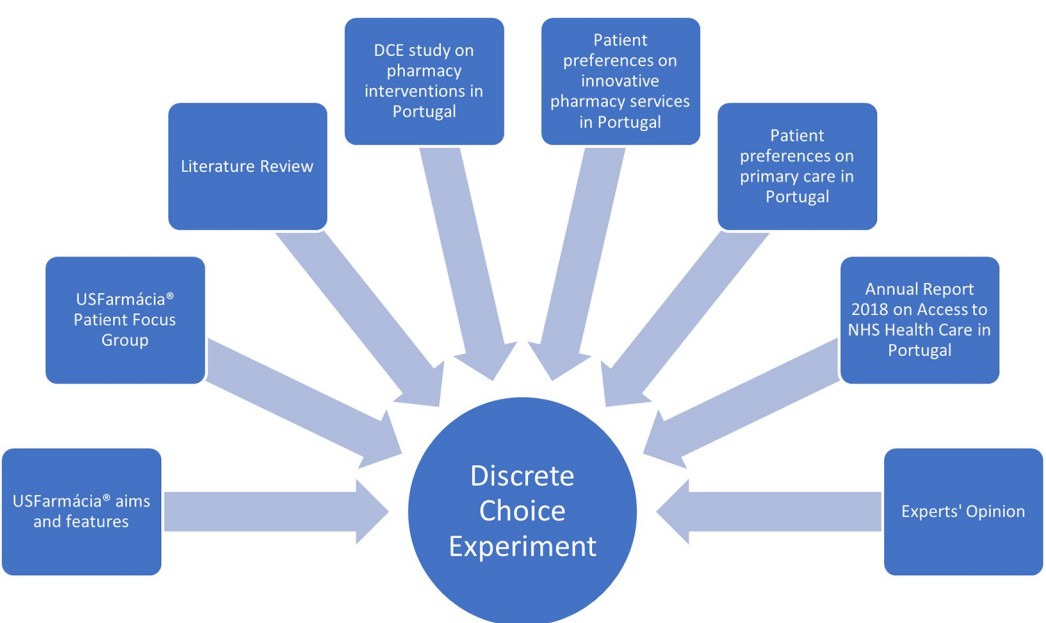

**Fig 1. Sources for attributes and levels of Discrete Choice Experiment (DCE).** NHS indicates National Health Service.

| Attributes | Levels |
|---|---|
| **#1 – Model of pharmacy intervention:** | 1) 5-minute at counter for a basic medication check of your prescription medicines<br>2) 15-minute in private office every 3 months for BP measurement and medication review of your medicines for high blood pressure and/or cholesterol<br>3) 30-minute in private office every 6 months for BP, total cholesterol, LDL, HDL and triglycerides measurement and medication review of ALL your medicines |
| **#2 - Integration with primary care** | 1) Weak – NO protocols pre-agreed with physicians and it is not possible to schedule medical appointment with your GP via pharmacy IT system<br>2) Partial – protocols pre-agreed with physicians BUT it is not possible to schedule medical appointment with your GP via pharmacy IT system<br>3) Full – protocols pre-agreed with physicians AND it is possible to schedule medical appointment with your GP via pharmacy IT system |
| **#3 - Waiting time to get requested medical appointment** | 1) 7 days (urgent) / 45 days (not urgent)<br>2) 48 hrs (urgent) / 30 days (not urgent)<br>3) Same day (urgent) / 15 days (not urgent) |
| **#4 - Chance of having a stroke in the next 5 years** | 1) Is the same<br>2) Is slightly lower<br>3) Is much lower |
| **#5 – Annual cost to the NHS** | 1) €0<br>2) €30<br>3) €51<br>4) €76 |

**Fig 2. Attributes and levels of discrete choice experiment.** BP indicates Blood Pressure; GP, General Practitioner; HDL, High-Density Lipoprotein; IT, Information Technology; LDL, Low-Density Lipoprotein; NHS, National Health Service.

## Model of pharmacy intervention

This included: time with the patient; degree of privacy; frequency of intervention; and type of intervention provided. We selected the usual care provided in most pharmacies (level 1) and two other realistic options (levels 2 and 3) illustrating collaborative care models based on the frequency of point-of-care measurements in trial and pharmacy-based medication reviews reimbursed in some countries [47]. Level 2 is a more frequent, yet less comprehensive intervention. Level 3 is a less frequent yet more comprehensive model of pharmacy intervention (the highest form of pharmacy intervention).

**Integration with primary care.** We selected usual care which is almost no integration (level 1) and two other options (levels 2 and 3) describing the degree of integration with primary care as planned for the trial intervention arm expressed in protocols with physicians and scheduling medical appointments directly from the pharmacy software. In level 2 there is partial integration. Level 3 corresponds to the full integration with primary care featured in the intervention arm of the collaborative trial.

**Waiting time to get requested medical appointment.** We defined level 3 as same day (urgent)/15 days (not urgent) and "intermediate" based on the Maximum Waiting Time for referrals to primary care defined in legislation [48]. This would be the "best" level for waiting time to pursue in future collaborative care. Level 2 was defined as 48 hrs. (urgent)/ 30 days (not urgent) options as a first step towards "best" adopted for the trial intervention arm. Intervention patients reported an average waiting time of 12 days. We defined possible usual care (level 1) as 7 days (urgent)/45 days (not urgent) based on: the Annual Report 2018 which states that 56% of urgent referrals to primary care were not performed on the same day [46]; control patients who reported an average waiting time of 41 days; a study in Portugal which stated that «across the focus groups, participants reported that waiting times for appointments with GPs could extend to several months» [45]; experts' opinion (community pharmacists consulted).

**Chance of having a stroke in 5 years.** Following Fletcher et al, who also studied hypertension management, we modified the chance of best treatment into chance of having a stroke in 5 years [29]. This attribute was selected as the reduction in risk is a proxy for the chance of receiving "best" intervention in hypertension and/or hyperlipidemia management. Level 1 corresponds to usual care (no change). In level 2, the chance of stroke is slightly lower as a result of some collaborative care. In level 3, the chance of stroke is much lower as a consequence of the desired highest level of collaborative care.

**Annual cost to the NHS.** The value can be studied from the perspective of WTP (costs defined as extra payment), the maximum amount a person would be willing to offer for a good, or of WTA (costs defined as discount or compensation), the minimum monetary amount required for an individual to forgo some good, or to bear some harm [49,50]. WTA measures the amount of money that is required *after* the change to make a respondent's level of utility the same as before the change [50]. In addition, as the NHS pays for health care in Portugal and the rationale is to have this intervention reimbursed, it would be unrealistic to ask patients for out-of-pocket payments to express their WTP. A review of DCEs described difficulty in determining the absolute value of a new intervention using WTP, particularly, in European health systems where patients pay little for health services [51]. Hence, we framed cost as annual cost to the NHS (WTA), as in Fletcher et al [29].

We selected usual care at €0 (level 1). The other three cost levels reflect varying amounts of reimbursement used for medicines in Portugal (100%, 69%, and 40%) applied to the cost of pharmacists' time for either 15 minutes 4 times a year or 30 minutes twice a year plus the cost of point-of-care measurements.

We used a pharmacist cost of €0.75 per minute based on: 2016 operating costs [52]; 2018 average operating hours per week [53] and 2018 average minimum pharmacist salary [54] for Portuguese pharmacies. This is similar to €0.74 to €0.78 per minute from Gregório et al's study [55].

We used cost of point-of-care measurements as valued by the Pharmacy Customer Loyalty Program Saúda® in 2018 for points redeem (price-proxy) since there are no fixed market prices for these services.

## Study design and questionnaire

The DCE contains choice sets of pharmacy-based hypertension and/or hyperlipidemia management models. The maximum theoretical number of possible combinations of the levels and attributes is 324 ($3^4 * 4^1$) using full factorial design. In a pairwise DCE with two alternative profiles, the total number of choice sets would then be 52,326 (324*323/2), which is not feasible to use. An experimental orthogonal fractional factorial design was generated in R (version 3.5.1) AlgDesign package statistical software to reduce the number of choice sets to 36.

The orthogonal design was preferred to minimize statistical error despite acknowledging the potential risk of including illogical combinations and becoming confusing for patients. Relative efficiency was measured using D-efficiency and the design had 93% efficiency.

These 36 choice sets were allocated into nine blocks (questionnaires). We limited each to 4 choice sets per patient. A clinical scenario preceding choice sets was included to provide context to the decision-making. For each choice set, patients were asked to elicit their preferred choice of service from two alternative profiles (A or B). We used a non-labeled design. We did not include an opt-out alternative. An example of a choice set is presented in **Fig 3**.

We performed a pretest in similar patients (n = 10). Results were used to inform two changes to the final version: levels of attribute "chance of having a stroke in 5 years" changed from probabilities to qualitative, and patients were asked to keep a pen and paper. The final survey instrument can be found in the **S1 Appendix**.

## Data collection

**DCE survey.** We performed the sample size calculation for our 6-month effectiveness trial resulting in a required minimum sample size of 322 hypertensive patients, considering a 20% drop-out rate, 80% power, and a 5% significance level to detect a change in our desired outcome. We also estimated the required minimum sample size for our DCE.

For sample size calculation, we took note of the "rule of thumb" method of Johnson and Orme for DCEs [56,57] which suggests that the sample size required for the main effects depends on the number of choice tasks (t), the number of alternatives (a), and the number of analysis cells (c) according to the following equation:

$$N > 500c/(t \times a)$$

Considering the main effects, we would require a minimum of 250 patients. However, we had to use available 122 trial patients who were recruited from trial patients who had replied to the 12-month telephone survey and reconfirmed consent.

Each intervention and control patient was randomly assigned to one of the nine versions of the questionnaire. The DCE survey was administered as a trial exit Computer-Assisted Telephone Interview (CATI) structured around a script. CATI had already been used in the trial to collect other patient-reported data. The database was validated prior to analysis. All interviewers were provided prior training to ensure compliance with data collection instructions, General Data Protection Regulation (GDPR), and ethics, and to maximize approach

| | ATTRIBUTE | SERVICE A | SERVICE B |
|---|---|---|---|
| 1 | Model of pharmacy intervention | 5-minute at counter for a basic medication check of your prescription medicines. | 15-minute in private office every 3 months for BP measurement and medication review of your medicines for high blood pressure and/or cholesterol |
| 2 | Integration with primary care | Poor – NO protocols pre-agreed with physicians and it is not possible to schedule medical appointment with your GP via pharmacy IT system | Full – protocols pre-agreed with physicians AND it is possible to schedule medical appointment with your GP via pharmacy IT system |
| 3 | Waiting time to get requested medical appointment | 48 hrs (urgent) / 30 days (not urgent) | 7 days (urgent) / 45 days (not urgent) |
| 4 | Chance of having a stroke in the next 5 years | Is much lower | Is slightly lower |
| 5 | Annual cost to the NHS | €76 | €51 |
| | | | |
| | WHICH SERVICE A OR B WOULD YOU CHOOSE? | ☐ | ☐ |

**Fig 3. Example of a choice set.** BP indicates Blood Pressure; GP, General Practitioner; IT, Information Technology; NHS, National Health Service.

standardization. In each choice set, interviewers were instructed to present each alternative with its full set of attributes, first alternative A, then B, and to repeat it if necessary.

The survey was conducted between 5 and 18 February 2020. Feedback from the first interviews was provided to the research team. We planned for up to eight tentative calls to minimize the loss of patients. Calls were spread over various days and hours to maximize success. When patients answered the call, but the time was inconvenient, the call was re-arranged.

**Patient characteristics and other variables.** Patient demographics, socioeconomics, clinical, waiting time for medical appointments, and cost data had already been collected alongside the trial. Income status used monthly equivalent income per person [58]; medication classes were defined as per the International Consortium for Health Outcomes Measurement (ICHOM) [59]; comorbidities used the Rx-Risk Comorbidity Index [60].

For the DCE survey, we collected 3 additional variables: "patient has GP", "patient pays NHS user charge", and "patient has health insurance or health sub-system".

## Data analysis

Demographics and case-mix variables were summarized using descriptive statistics. We assessed the quality of responses by evaluating the internal validity of the data and looking for respondents who always chose the alternative profile with the best level of one attribute in all 4 choice sets because preferences that are dominated by a single attribute can bias model estimation.

DCE data were analyzed in IBM SPSS Statistics (v.26) by conditional logit regression using the COXREG procedure. The primary choice for this model assumed intervention trial patients shared many baseline characteristics since they all had to meet pre-defined inclusion

criteria, and they had experienced the same standardized patient care collaborative intervention, thereby reducing the risk of preference heterogeneity and variance across choice tasks [43]. In addition, as this was a trial exit interview, we had to work with available patients and did not have a large enough sample size to test alternative models.

Dummy variables were used to analyze categorical attributes with reference levels identified in Table 2. The magnitude of the regression coefficients (β) expressed the degree of preference for each of the attribute levels, in which the greater the coefficient, the more that attribute level was preferred. Confidence intervals (CI) of regression coefficients were estimated using bootstrap with 1,000 replications.

Trade-offs between attributes can be demonstrated by using utility scores (V), calculated by using the following equation:

$$V = \beta_{model} + \beta_{integration} + \beta_{\text{wait time}} + \beta_{\text{stroke}} + \beta_{\text{cost}} cost$$

The model fit was assessed by log-likelihood and the chi-square value for the difference between the full and null models.

We calculated the relative importance of each attribute which is the ratio of each attribute's utility range (difference between the highest and lowest utility value) to the sum of the utility range for all attributes.

We used the ratio of the coefficient for the attribute of interest ($\beta_x$) to the cost coefficient ($\beta_{cost}$) to calculate WTA for marginal changes in attributes (change from reference case to another attribute level). The total WTA for a particular model configuration was calculated by taking the sum of the WTAs of each attribute.

CIs of marginal WTA for each attribute level were estimated using the parametric bootstrap method with 1,000 replications, as suggested by Krinsky and Robb and quoted by Hole [61].

We calculated the aggregate WTA annual cost to the NHS for a policy change from the reference case (usual care) to the most preferred scenario in post-estimation welfare analysis using the compensating variation methodology [62].

The welfare change in moving from the reference case (usual care) to the "best" case scenario (ideal intervention) was incorporated into a CBA framework.

Baseline and 6-month trial costs are reported elsewhere [41]. We have used the estimated annual incremental costs to perform CBA.

Although we estimated preferences from the perspective of both intervention and control patients, mean WTA was compared to estimated annual incremental costs using Net Benefit (WTA—costs) from the perspective of intervention patients. We chose this as it could be argued that the relevant WTP (or WTA) should come from the individuals who were part of the intervention group because the control group may lack a firm understanding of the collaborative care model [28].

## Results

From the initial set of 143 patients, 8 refused and 13 could not be reached after eight phone calls. A total of 122 completed the DCE survey. Patients replied *yes* when prompted if they found the survey *interesting* (66%), *hard to understand* (14%), *not realistic* (11%), and *too long* (5%). Between 0 and 14% of respondents chose the alternative profile with the best level of one attribute always in all 4 choice sets: 9 (7%) patients in model of intervention; 5 (4%) patients in integration with primary care; 17 (14%) patients in waiting time to get requested medical appointment; and 3 (2%) patients in annual cost to the NHS. Since these are few patients and preferences do not seem to be dominated by a single attribute, we included these patients in the analysis.

DCE patient characteristics are shown in **Table 1.**

**Table 1. DCE patient demographics and case-mix at baseline.**

| Demographics and Case Mix at Baseline | Intervention (n = 78) | Control (n = 44) | P-Value for Difference |
|---|---|---|---|
| **Gender** (NR = 0) (C) | | | 0.7108 |
| Female | 47 (60.3%) | 25 (56.8%) | |
| Male | 31 (39.7%) | 19 (43.2%) | |
| **Age, years (mean ± SD)** (NR = 4) (C) | 65.9 (10.9) | 65.0 (10.4) | 0.3887 |
| **Education** (NR = 10) (C) | | | |
| No. years education, mean (SD) | 9.1 (4.4) | 7.8 (4.8) | 0.0924 |
| Education ≤ Elementary School 3rd cycle (current 9th grade / former 5th grade / technical schools) | 43 (59.7) | 28 (70.0) | 0.2793 |
| **Employment status** (NR = 9) (C) | | | |
| Retired/pensioner + permanently disabled + unemployed + household tasks | 54 (75.0%) | 29 (70.7%) | 0.6213 |
| **Income** (NR = 33) | | | |
| Approx. monthly equivalent income per person* (= household income average threshold/no. of equalized individuals in household) in € (SD) | 824.23€ (555.48 €) | 582.42€ (388.79 €) | 0.0150 |
| Approx. monthly household income (C) (= household income average threshold) in € (SD) | 1256.14€ (870.35 €) | 939.39€ (661.88 €) | 0.0584 |
| ≤ €501.20 (n, %) | 17 (30.4%) | 16 (48.5%) | 0.0872 |
| **Municipality Purchasing Power Index (IPCC)** | 95,23 | 92,5 | |
| **Smoking status (n, %)** (NR = 9) (C) | | | |
| Smoker (Y) | 6 (8.2%) | 4 (10.0%) | 0.7407 |
| **BMI (mean kg/m$^2$ ± SD)** (NR = 13) (C) | 27.1 (4.5) | 28.5 (4.4) | 0.1226 |
| **Comorbidities** (NR = 5)** | | | |
| No. comorbidities per patient (mean ± SD) | 1.9 (1.5) | 2.4 (2.1) | 0.3002 |
| **No. regular medicines per patient** (B)*** | | | |
| Mean, (SD) | 4.4 (2.4) | 4.9 (3.2) | 0.7034 |
| **Patients on** (A, B, C) | | | 0.2604 |
| Antihypertensive medication (n, %) (NR = 0) | 23 (42.3%) | 13 (29.5) | |
| Lipid-lowering medication (n, %) (NR = 0) | 22 (28.2%) | 7 (15.9%) | |
| Antihypertensive lipid-lowering medication (n, %) (NR = 0) | 33 (42.3%) | 24 (54.5%) | |
| **Number of years since onset (mean ± SD)** (C) | | | |
| Antihypertensive medication | 5.9 (6.4) | 6.7 (7.2) | 1.0000 |
| Lipid-lowering medication | 4.4 (4.5) | 6.4 (7.4) | 0.1899 |
| **Antihypertensive medication** (B) | | | |
| No. antihypertensive medicines per patient (mean ± SD) | 1.5 (0.7) | 1.8 (1.0) | 0.3334 |
| **Patients** (D) | | | |
| Without General Practitioner (n, %) (NR = 1) | 0 (0.0%) | 3 (7.0%) | 0.0429 |
| Paying NHS user charge (n, %) (NR = 0) | 62 (79.5%) | 27 (61.4%) | 0.0305 |
| With health insurance / health sub-system (n, %) (NR = 0) | 32 (41.0%) | 19 (43.2%) | 0.8166 |
| **Number of days (mean ± SD)** (C) | | | |
| Waiting Timing for appointments (NR = 32) | 12.1 (12.4) | 41.2 (55.9) | 0.0067 |

A: Pharmacy dispensing software; B: Primary care software; BMI: Body Mass Index; C: Telephone baseline survey; D: DCE survey; NHS: National Health Service; NR: Non-Respondents.

* Derived from monthly household income.

** Derived from prescribed medicines using Rx-Risk Comorbidity Index.

*** Number of medicines equals number of different International Non-Proprietary Names.

Intervention patients have a significantly higher monthly equivalent income, a slightly higher proportion pays the NHS user charge, and lower waiting time for medical appointments vs. control patients (P<0.05) (**Table 1**).

Monthly equivalent income and payment of the NHS user charge were not covariates in the regression model. Waiting time for medical appointments was an attribute in itself.

Results from the regression analysis and WTA estimates are presented in **Table 2**.

All coefficients of attributes *waiting time to get medical appointment*, *model of pharmacy intervention*, and *integration with primary care* were positive, indicating that collaborative care options were preferred to reference levels, confirming theoretical validity. At nine degrees of freedom, chi-square values for the difference of log-likelihood between the full and null models are all statistically significant at p < 0.001, suggesting the model is a significant predictor.

Waiting time to get a medical appointment was a significant driver of preferences, with all patients preferring waiting time on the same day (urgent) and within 15 days (non-urgent). This preference was stronger in control patients.

Patients also preferred: 30-minute in private office every 6 months for BP, total cholesterol, LDL, HDL, and triglycerides (TG) measurement and medication review of all their medicines, and full integration with protocols with physicians and ability to schedule a medical appointment via pharmacy software. These preferences were stronger for intervention patients.

Since the coefficients of the chance of stroke and the annual cost to the NHS were not statistically significant, we just calculated the relative importance of the remaining attributes. Again, we confirmed waiting time to get a medical appointment had the highest relative importance (57%) followed by the model of pharmacy intervention (25%) and integration with primary care (18%).

Although the coefficients of the chance of stroke and the annual cost to the NHS were not statistically significant, trade-offs between the remaining attributes were calculated to determine the WTA cost to the NHS (**Table 2**) for intervention patients.

All else equal, WTA annual cost to the NHS for waiting time to get the requested medical appointment on the same day (urgent) and within 15 days (non-urgent) was €349.67.

WTA annual cost to the NHS for 30 minutes in private office every 6 months for BP, total cholesterol, LDL, HDL, and TG measurement and medication review of all medicines was €316.00.

WTA annual cost to the NHS for protocols with physicians and the ability to schedule a medical appointment with the GP via pharmacy software was €211.33.

The total marginal WTA annual cost to the NHS for the most preferential scenario was €877 (**Table 3**).

Combining WTA values with incremental costs of €88.80 from previous research [41], the net benefit per patient is €788.20 and represents the monetary value of intervention patients' welfare surplus for this collaborative model.

## Discussion

### Summary of key findings

Waiting time to get the requested medical appointment was a significant driver of preferences. Preference was stronger for control patients probably because they reported a higher average waiting time.

The model of pharmacy intervention and integration with primary care also played an important part in decision-making. These preferences were stronger in intervention patients probably because they experienced similar features in the trial.

Our results are consistent with international DCE patient enhanced care pharmacy studies where patients express a preference for a shorter waiting time [15,16,18,25,26,28,33,34,36], a

**Table 2. Regression analysis and Willingness to Accept (WTA).**

| Attribute and levels | All | | | Intervention | | | Control | | |
|---|---|---|---|---|---|---|---|---|---|
| | Regression coefficient (β) (95% CI) | P-Value | WTA (€) (95% CI) | Regression coefficient (β) (95% CI) | P-Value | WTA (€) (95% CI) | Regression coefficient (β) (95% CI) | P-Value | WTA (€) (95% CI) |
| **Model of pharmacy intervention:** | | | | | | | | | |
| 5-minute at counter for a basic medication check of your prescription medicines (a) | [0] | | - | [0] | | - | [0] | | - |
| 15-minute in private office every 3 months for BP and medication review of your medicines for high BP and/or TC | 0.551 (0.290–0.844) | 0.003 | 275.50 (-1964.7–2914.8) | 0.605 (0.287–1.036) | 0.009 | 201.67 (-1706.4–2234.1) | 0.481 (-0.004–1.097) | 0.119 | 481.00 (-1691.0–1630.3) |
| 30-minute in private office every 6 months for BP, TC, LDL, HDL, and TG and medication review of ALL your medicines | 0.888 (0.586–1.264) | 0.000 | 444.00 (-3495.8–3331.5) | 0.948 (0.547–1.458) | 0.000 | 316.00 (-2682.7–2816.3) | 0.792 (0.171–1.543) | 0.026 | 792.00 (-2846.2–2694.7) |
| **Integration with primary care:** | | | | | | | | | |
| Weak–NO protocols pre-agreed with physicians, not possible to schedule medical appointment with GP via pharmacy IT (a) | [0] | | - | [0] | | - | [0] | | - |
| Partial–protocols pre-agreed with physicians, not possible to schedule medical appointment with GP via pharmacy IT | 0.376 (0.116–0.656) | 0.023 | 188.00 (-1614.8–1025.5) | 0.417 (0.100–0.781) | 0.043 | 139.00 (-1160.1–1063.7) | 0.308 (-0.199–0.902) | 0.269 | 308.00 (-2490.5–841.2) |
| Full–protocols pre-agreed with physicians, possible to schedule medical appointment with GP via pharmacy IT | 0.622 (0.327–0.963) | 0.001 | 311.00 (-2848.3–2790.5) | 0.634 (0.257–1.083) | 0.007 | 211.33 (-2173.2–2612.5) | 0.617 (0.044–1.285) | 0.055 | 617.00 (-3059.2–1425.9) |
| **Waiting time to get medical appointment:** | | | | | | | | | |
| 7 days (urgent) / 45 days (not urgent) (a) | [0] | | - | [0] | | - | [0] | | - |
| 48 hrs. (urgent) / 30 days (not urgent) | 0.381 (0.121–0.690) | 0.022 | 190.50 (-1577.5–1551.3) | 0.267 (-0.067–0.675) | 0.199 | 89.00 (-1082.7–1033.0) | 0.585 (0.148–1.200) | 0.038 | 585.00 (-3189.1–1395.5) |
| Same day (urgent) / 15 days (not urgent) | 1.150 (0.893–1.474) | 0.000 | 575.00 (-4421.6–4419.4) | 1.049 (0.756–1.489) | 0.000 | 349.67 (-3162.9–2923.4) | 1.349 (0.944–2.002) | 0.000 | 1349.00 (-3500.2–4191.6) |
| **Chance of having a stroke in 5 years:** | | | | | | | | | |
| Is the same (a) | [0] | | - | [0] | | - | [0] | | - |
| Is slightly lower | 0.121 (-0.263–0.500) | 0.589 | - | 0.181 (-0.281–0.661) | 0.516 | - | 0.022 (-0.732–0.810) | 0.954 | - |
| Is much lower | 0.067 (-0.234–0.366) | 0.712 | - | 0.232 (-0.136–0.629) | 0.309 | - | -0.220 (-0.723–0.322) | 0.463 | - |
| **Annual cost to NHS (€):** | 0.002 (-0.003–0.008) | 0.510 | - | 0.003 (-0.004–0.010) | 0.514 | - | 0.001 (-0.009–0.011) | 0.805 | - |
| Log-likelihood ratio | 3113.86 | | | 1712.88 | | | 763.053 | | |
| $\chi^2$ (9) | 101.28 | | | 63.66 | | | 40.303 | | |
| Number of individuals (observations) | 122 (976) | | | 78 (624) | | | 44 (352) | | |

(a) Reference category.

BP: Blood Pressure; GP: General Practitioner; HDL: High-Density Lipoprotein; IT: Information Technology; LDL: Low-Density Lipoprotein; NHS: National Health Service; TC: Total Cholesterol; TG: Triglycerides; WTA: Willingness-To-Accept.

**Table 3. Marginal WTA between the most preferred scenario and the least preferred pharmacy service bundle scenario for intervention patients.**

| Attribute | Most preferred scenario | Least preferred scenario | Marginal WTA (€) |
|---|---|---|---|
| Model of pharmacy intervention | 30-minute in private office every 6 months for BP, TC, LDL, HDL, and TG measurement and medication review of ALL your medicines | 5-minute at counter for a basic medication check of your prescription medicines | 316.00 |
| Integration with primary care | Full–protocols pre-agreed with physicians AND it is possible to schedule a medical appointment with your GP via pharmacy IT system | Weak–NO protocols pre-agreed with physicians and it is not possible to schedule a medical appointment with your GP via pharmacy IT system | 211.33 |
| Waiting time to get requested medical appointment | Same day (urgent) / 15 days (not urgent) | 7 days (urgent) / 45 days (not urgent) | 349.67 |
| Total | | | 877.00 |

BP: Blood Pressure; GP: General Practitioner; HDL: High-Density Lipoprotein; IT: Information Technology; LDL: Low-Density Lipoprotein; TC: Total Cholesterol; TG: Triglycerides; WTA: Willingness-To-Accept.

similar model of pharmacy intervention [15,17,19,20,24,28,31,35,36], and the chance of best treatment [18,28,29]. There are still very few DCE studies published on pharmacy-led integrated or collaborative care with physicians, but these also confirm preferences for integration with physicians [27,28]. Almost all studies framed cost as WTP which makes it difficult to compare but cost is a very strong preference for patients in all these studies when considering an enhanced pharmacy care model or intervention. In addition, the only study that incorporated WTP in a CBA model and in trial patients who experienced an innovative medication-related service also demonstrated a positive Net Benefit [28].

Results are also consistent with previous research in Portugal where patients express a preference for a shorter waiting time to get a medical appointment [45]. Preferences for comprehensive models of pharmacy intervention and scheduled medical appointments in the pharmacy have also been established [44].

The chance of having a stroke in 5 years did not seem to contribute to decision-making. However, some patients may have not fully understood this attribute.

Since we were interested in understanding preferences for public health intervention in a scenario of reimbursement within a tax-financed (public) healthcare system, we framed cost as WTA cost to the NHS, but this may have contributed to an almost null sensitivity.

Finally, we sought to estimate the total marginal WTA annual cost to the NHS for the most preferential scenario and the Net Benefit Value. We used the perspective of intervention patients, as argued in previous research [28], to provide a more conservative valuation, as loss aversion was lower than experienced by control patients.

## Strengths and limitations

**Limitations.** The choices patients make in real life may not always be the same as the choices they make in a hypothetical DCE. This is known as hypothetical bias and is a general limitation of DCEs.

DCE is a complex method that works best through face-to-face interviews in older or less literate populations. We used a telephone survey as this was used to collect other patient-reported trial data. We tried to minimize patient burden and risk of poor recall of all variables in a telephone interview with no visualization by 1) keeping choice sets to four; 2) requesting patients to keep a pen and paper handy following recommendations from the pretest; 3) using CATI technology and trained interviewers; 4) providing detailed instruction to the interviewer in the survey before each scenario to standardize the approach and to allow for repetition

before collecting the patient's response. The self-administered online survey was not feasible due to the age and literacy level of our population. It could be interesting to explore online surveys in older populations through SMS or WhatsApp® in the future.

We did not include an opt-out alternative, (e.g. patients' current pharmacy service features), but we did not have a large enough sample size to account for the loss of preference data. Choosing an opt-out alternative does indicate the relative preference of patients who do not wish to choose between available alternatives, but an opt-out alternative may have implications as patients need to be censored in preference assessment. Hence, a large enough sample size is required for assessing preferences on attributes and levels to account for the loss of patients who choose an opt-out alternative.

We could not reach the minimum sample size required according to Johnson and Orme for DCEs [56,57]. We acknowledge the sample is small, but we did plan for an adequate sample size. Hence, the sample size estimate for the effectiveness trial would also be more than sufficient for our trial exit DCE survey. Recruitment efforts aimed, therefore, at achieving 322 hypertensive patients. However, this was a pragmatic controlled trial to assess effectiveness under real-world conditions where not all variables can be controlled by the researcher despite the planning. And, in practice, the research team faced unexpected external challenges which affected recruitment and prevented it from reaching the desired sample size. We reported this in detail in the research paper on the effectiveness trial [40].

The survey version used in the pretest included probabilities in the risk reduction attribute and this was not well understood which has been reported in previous research [63]. Despite the attempt to improve the final version, it probably remained an issue. We acknowledge that having an attribute that may have not been fully understood has the potential for patients to assign additional value to other variables which may affect the WTA of each attribute but less likely the relative importance of each attribute or the sum of all attribute WTA used to determine CBA.

We framed cost as WTA cost to the NHS because, in health care, WTA is more appropriate for potential "losers", e.g. individuals that may experience loss aversion as an intrinsic feeling of loss toward a new intervention, or when the health care intervention has already completed and changes in health state may have been experienced or perceived [50,64] which was the case of our DCE patients. WTA has also been studied to a greater degree in universal health-care environments [65]. However, the approach is not without challenges. Future research could use both WTA and WTP with cost framed as a public tax increase.

We did not include some patient characteristics such as interacting covariates in the regression model due to a high proportion of missing data, which would reduce the sample size. Hence, we may have missed the effect of individual heterogeneity in preferences. Yet, we assumed low heterogeneity for trial patients where inclusion criteria apply. Furthermore, although an income effect occurs in WTP because payment is constrained by income, it does not occur in WTA [50]. This, to some extent, may also apply to the payment of NHS user charges. Age, gender, and education did not differ significantly between groups.

We did not test alternative regression models, such as Random-Parameters Logit (RPL), Hierarchical Bayes, or Latent Class Models (LCM) to allow for unobserved heterogeneity and compare the consistency of results, as we did not have a large enough sample size [43].

It is also likely that chronic disease patients' perception of pharmacy services has evolved since the time this study was conducted. Community pharmacists have undertaken relevant direct patient care interventions during the pandemic in response to the extended closure of NHS face-to-face GP appointments [66]. As such, chronic patients may now have a better awareness of the patient care services that can be provided in pharmacies, as also reported by other authors [35].

These limitations preclude the generalization of our findings but offer valuable lessons from a real-world trial for future DCE studies in pharmacy-based public health interventions.

**Strengths.** Strengths of this study include various data sources to inform attributes and levels; relevant key attributes as distinct as possible from one another; the use of choice sets with random pairing; orthogonal design to minimize correlation among attributes and levels; non-labeled design to minimize selection bias; and analysis of uncertainty.

This study enables the understanding of the strength of preferences and trade-offs between attributes and provides welfare estimates, despite uncertainty, for pharmacy services where market choices are constrained by regulatory factors and there is potential for reimbursement. Patients must choose an alternative, trade-offs can be measured, and WTA can be estimated. It further attempts to incorporate DCE into an economic evaluation using trial costs previously estimated. Finally, it offers a mechanism for patients to participate in decision-making and seeks to capture other benefits not captured in CEA or CUA.

## Conclusions

### Implications for policy and practice

To our knowledge, this is the first study conducted in Portugal and one of the few worldwide on disease management alongside a pharmacy-based controlled trial, incorporating DCE and WTA into an economic evaluation using CBA.

This study can be used to guide the design and implementation of pharmacy collaborative interventions with primary care with the potential for reimbursement for uncontrolled or at-risk chronic disease patients informed by patient preferences.

### Implications for research

Some findings emerged, warranting further investigation, namely high loss aversion by control patients who have not experienced any of the preferred features. We suggest future DCE studies in pharmacy services with the potential for reimbursement in universal health care settings to incorporate both WTP and WTA. Alternative econometric regression models such as RPL or LCM could be explored to allow for unobserved heterogeneity. Demonstrating the external validity of DCEs is also important to address in future research.

DCE is a promising yet challenging methodology in pharmacy practice research. Future real-world DCE studies conducted in community pharmacy may provide additional contributions to this research.

## Supporting information

**S1 Appendix. Questionnaire on patient preferences (version Q1).**
(PDF)

## Acknowledgments

We express our gratitude to all patients who participated in this study.

We would like to thank the following individuals: Prof. Miguel Gouveia at CATÓLICA-LISBON Business & Economics, for valuable methodological inputs before this study, pharmacists Mónica Gomes, Susana Neves, and Wilson Godinho at Farmácia Estácio and Farmácia Estácio Xabregas, for performing the pretest, Rui Costa and Rita Garcia of Spirituc, for supervising DCE telephone survey interviews and data collection, Sónia Romano of CEFAR for assisting with Spirituc, Prof. Benjamin Fletcher, at the University of Oxford, UK, for sharing

the statistical methods of his study, Prof. Paula Veiga, at Universidade do Minho, for sharing the Final Report "What do patients want from primary care services?" *(Under an embargo period)* containing a DCE and WTP experiment in primary care with different statistical methods, Dr. Michela Tinelli, at the London School of Economics and Political Science, UK, for providing clarifications on cost data and net benefit estimate of her study, and António Teixeira Rodrigues for the support received from CEFAR researchers.

## Author Contributions

**Conceptualization:** Suzete Costa.

**Formal analysis:** Suzete Costa, José Guerreiro.

**Funding acquisition:** Suzete Costa.

**Investigation:** Suzete Costa.

**Methodology:** Suzete Costa, João Pereira.

**Project administration:** Suzete Costa.

**Resources:** Suzete Costa.

**Software:** Suzete Costa, José Guerreiro, Inês Teixeira.

**Supervision:** Dennis K. Helling, Céu Mateus, João Pereira.

**Validation:** José Guerreiro, Inês Teixeira.

**Visualization:** Suzete Costa.

**Writing – original draft:** Suzete Costa.

**Writing – review & editing:** Suzete Costa, José Guerreiro, Inês Teixeira, Dennis K. Helling, Céu Mateus, João Pereira.

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
