## [Decision Letter · Decision Letter 0]

27 Mar 2023

PONE-D-22-23923Patient preferences and cost-benefit of hypertension and hyperlipidemia collaborative management model between pharmacies and primary care in Portugal: A discrete choice experiment alongside a trial (USFarmácia®)PLOS ONE

Dear Dr. Costa,

Thank you for submitting your manuscript to PLOS ONE. After careful consideration, we feel that it has merit but does not fully meet PLOS ONE’s publication criteria as it currently stands. Therefore, we invite you to submit a revised version of the manuscript that addresses the points raised during the review process. One of the reviewer raises relatively minor comments, while the other submits more stringent critiques that require substantial effort to be addressed. My recommendation is to carefully read both reports and to carefully tackle each point raised, as we cannot provide any guarantee of publication at this stage.

We look forward to receiving your revised manuscript.

Kind regards,

Matteo Lippi Bruni, PhD

Academic Editor

PLOS ONE

Journal Requirements:

2. Thank you for providing the following Funding Statement: 

“I have read the journal's policy and the authors of this manuscript have the following competing interests: SC was employed by the Associação Nacional das Farmácias (ANF) [National Association of Pharmacies] at the time of trial onset and recruitment preceding this research until 2019. JG and IT are employed by the ANF. DKH has given talks on US innovative pharmacist-led collaborative interventions in Portugal for which travel and accommodation costs have been reimbursed by ANF. CM and JP declare that they have no known conflicts of interest.”

We note that one or more of the authors is affiliated with the funding organization, indicating the funder may have had some role in the design, data collection, analysis or preparation of your manuscript for publication; in other words, the funder played an indirect role through the participation of the co-authors.

If the funding organization did not play a role in the study design, data collection and analysis, decision to publish, or preparation of the manuscript and only provided financial support in the form of authors' salaries and/or research materials, please review your statements relating to the author contributions, and ensure you have specifically and accurately indicated the role(s) that these authors had in your study in the Author Contributions section of the online submission form. Please make any necessary amendments directly within this section of the online submission form.  Please also update your Funding Statement to include the following statement: “The funder provided support in the form of salaries for authors [insert relevant initials], but did not have any additional role in the study design, data collection and analysis, decision to publish, or preparation of the manuscript. The specific roles of these authors are articulated in the ‘author contributions’ section.”

If the funding organization did have an additional role, please state and explain that role within your Funding Statement.

Please also provide an updated Competing Interests Statement declaring this commercial affiliation along with any other relevant declarations relating to employment, consultancy, patents, products in development, or marketed products, etc. 

3. Please ensure that you include a title page within your main document. You should list all authors and all affiliations as per our author instructions and clearly indicate the corresponding author.

Reviewers' comments:

Reviewer's Responses to Questions

**Comments to the Author**

1. Is the manuscript technically sound, and do the data support the conclusions?

Reviewer #1: Yes

Reviewer #2: Partly

2. Has the statistical analysis been performed appropriately and rigorously? 

Reviewer #1: Yes

Reviewer #2: Yes

3. Have the authors made all data underlying the findings in their manuscript fully available?

Reviewer #1: No

Reviewer #2: Yes

4. Is the manuscript presented in an intelligible fashion and written in standard English?

Reviewer #1: Yes

Reviewer #2: Yes

5. Review Comments to the Author

Reviewer #1: Thanks for the opportunity to review your article, Patient preferences and cost-benefit of hypertension and hyperlipidemia collaborative management model between pharmacies and primary care in Portugal: A discrete choice experiment alongside a trial (USFarmácia®). This is a well written article and I have minor comments for clarification:

1- Further explanation about the difference between the WTP/WTA model in DCEs should be clarified (and specific wording used for this experiment) in light also of the literature https://doi.org/10.1111/j.1524-4733.2008.00340.x.

2- In the discussion the authors mentioned that sample size calculations were not provided as they had to rely on a predefined number of patients enrolled in the trial. Still, it would be important to understand whether the sample size available was suitable for the DCE exercise (and calculations should be reported in the methods section).

3- The choice of the regression model is not clarified. I think the readers would appreciate to know whether alternative models have been tested (and the selection criteria applied to find the model with the best fit).

Reviewer #2: Please see attached. While a conditional logit is appropriate for DCE surveys, I have concerns related to design and sample size. Additionally, I have concerns related to how the limitations of the study may affect the results.

6. PLOS authors have the option to publish the peer review history of their article (what does this mean?). If published, this will include your full peer review and any attached files.

Reviewer #1: No

Reviewer #2: No

---

## [Author Response · Author response to Decision Letter 0]

31 May 2023

We have uploaded a response to reviewers file.

---

## [Decision Letter · Decision Letter 1]

17 Jul 2023

PONE-D-22-23923R1Patient preferences and cost-benefit of hypertension and hyperlipidemia collaborative management model between pharmacies and primary care in Portugal: A discrete choice experiment alongside a trial (USFarmácia®)PLOS ONE

Dear Dr. Costa,

Thank you for submitting your manuscript to PLOS ONE. After careful consideration, we feel that it has merit but does not fully meet PLOS ONE’s publication criteria as it currently stands. Therefore, we invite you to submit a revised version of the manuscript that addresses the points raised during the review process.

We look forward to receiving your revised manuscript.

Kind regards,

Matteo Lippi Bruni, PhD

Academic Editor

PLOS ONE

Journal Requirements:

**Additional Editor Comments: **

Both reviewers are satisfied with the way you have addressed the comments they raised during the first round. Only one minor comment remains, which you find specified in detail in one of the reports. Once this point is resolved, I will proceed with the final decision without sending the article back to the reviewers again.

Reviewers' comments:

Reviewer's Responses to Questions

**Comments to the Author**

1. If the authors have adequately addressed your comments raised in a previous round of review and you feel that this manuscript is now acceptable for publication, you may indicate that here to bypass the “Comments to the Author” section, enter your conflict of interest statement in the “Confidential to Editor” section, and submit your "Accept" recommendation.

Reviewer #1: All comments have been addressed

Reviewer #2: All comments have been addressed

2. Is the manuscript technically sound, and do the data support the conclusions?

Reviewer #1: Yes

Reviewer #2: Partly

3. Has the statistical analysis been performed appropriately and rigorously? 

Reviewer #1: Yes

Reviewer #2: Yes

4. Have the authors made all data underlying the findings in their manuscript fully available?

Reviewer #1: (No Response)

Reviewer #2: Yes

5. Is the manuscript presented in an intelligible fashion and written in standard English?

Reviewer #1: Yes

Reviewer #2: Yes

6. Review Comments to the Author

Reviewer #1: I reviewed the revised version of the paper and am satisfied with the changes performed to the manuscript apart from one last comment. Page 18, the aggregate willingness to accept monetary amounts in response to policy changes when changing from the least preferred scenario to the most preferred scenario can be calculated in post-estimation welfare analysis using the compensating variation methodology. This should be referenced in the paper.

Reviewer #2: While I still have concerns regarding the sample size and attribute interpretation by participants, I believe the discussion of these limitations is sufficient to warrant publication of this work. Thank you for addressing my initial comments which were extensive.

7. PLOS authors have the option to publish the peer review history of their article (what does this mean?). If published, this will include your full peer review and any attached files.

Reviewer #1: No

Reviewer #2: No

---

## [Author Response · Author response to Decision Letter 1]

2 Sep 2023

We have uploaded a response to reviewers (round 2) together with revised manuscript with track changes and clean manuscript.

We have uploaded figures after PACE.

We have also corrected 2 affiliations to correct designation and added a third affiliation for the first author (marked in YELLOW)

We have added 2 new references (marked in YELLOW).

We have reviewed all references to conform to journal requirements and added links where available.

---

## [Editor Report · Decision Letter 2]

18 Sep 2023

Patient preferences and cost-benefit of hypertension and hyperlipidemia collaborative management model between pharmacies and primary care in Portugal: A discrete choice experiment alongside a trial (USFarmácia®)

PONE-D-22-23923R2

Dear Dr. Costa,

We’re pleased to inform you that your manuscript has been judged scientifically suitable for publication and will be formally accepted for publication once it meets all outstanding technical requirements.

Kind regards,

Matteo Lippi Bruni, PhD

Academic Editor

PLOS ONE
---

## [Editor Report · Acceptance letter]

25 Sep 2023

PONE-D-22-23923R2 

Patient preferences and cost-benefit of hypertension and hyperlipidemia collaborative management model between pharmacies and primary care in Portugal: A discrete choice experiment alongside a trial (USFarmácia®) 

Dear Dr. Costa:

I'm pleased to inform you that your manuscript has been deemed suitable for publication in PLOS ONE. Congratulations! Your manuscript is now with our production department. 

Kind regards, 

on behalf of

Dr. Matteo Lippi Bruni 

Academic Editor

PLOS ONE